# Evaluation of the Mechanisms Involved in the Development of Bladder Toxicity following Exposure to Occupational Bladder Cancer Causative Chemicals Using DNA Adductome Analysis

**DOI:** 10.3390/biom14010036

**Published:** 2023-12-26

**Authors:** Shugo Suzuki, Min Gi, Masami Komiya, Asuka Obikane, Arpamas Vachiraarunwong, Masaki Fujioka, Anna Kakehashi, Yukari Totsuka, Hideki Wanibuchi

**Affiliations:** 1Department of Molecular Pathology, Graduate School of Medicine, Osaka Metropolitan University, 1-4-3 Asahi-machi, Abeno-ku, Osaka 545-8585, Japan; shugo@omu.ac.jp (S.S.); mwei@omu.ac.jp (M.G.); arpamas.vachi@omu.ac.jp (A.V.); tajiri.fujioka@omu.ac.jp (M.F.); anna-k@omu.ac.jp (A.K.); 2Department of Environmental Risk Assessment, Graduate School of Medicine, Osaka Metropolitan University, 1-4-3 Asahi-machi, Abeno-ku, Osaka 545-8585, Japan; 3Laboratory of Environmental Toxicology and Carcinogenesis, School of Pharmacy, Nihon University, Chiba 274-8555, Japan; sakano.masami@nihon-u.ac.jp; 4Division of Cancer Pathophysiology, National Cancer Center Research Institute, Tokyo 104-0045, Japan; g22sm003@yamanashi.ac.jp; 5Department of Biochemistry, University of Yamanashi, Shimokato 1110, Chuo 409-3898, Japan

**Keywords:** *o*-toluidine, bladder, DNA adductome, oxidative stress

## Abstract

Occupational exposure to aromatic amines (AAs) is an important risk factor for urinary bladder cancer. This study aimed to evaluate the toxicity of AAs and analyze the carcinogenic mechanisms in rat bladder by comprehensive analysis of DNA adducts (DNA adductome). DNA was extracted from the bladder epithelia of rats treated with AAs, including acetoacet-*o*-toluidine (AAOT) and *o*-toluidine (OTD), and adductome analysis was performed. Principal component analysis–discriminant analysis revealed that OTD and AAOT observed in urinary bladder hyperplasia could be clearly separated from the controls and other AAs. After confirming the intensity of each adduct, four adducts were screened as having characteristics of the OTD/AAOT treatment. Comparing with the in-house DNA adduct database, three of four candidates were identified as oxidative DNA adducts, including 8-OH-dG, based on mass fragmentation together with high-resolution accurate mass (HRAM) spectrometry data. Therefore, findings suggested that oxidative stress may be involved in the toxicity of rat bladder epithelium exposed to AAs. Consequently, the administration of apocynin, an inhibitor of nicotinamide adenine dinucleotide phosphate oxidase, in six-week-old rats fed with 0.6% OTD in their diet resulted in simple hyperplastic lesions in the bladder that were suppressed by apocynin. The labeling indices of Ki67, γ-H2AX, and 8-OHdG were significantly decreased in an apocynin concentration-dependent manner. These findings indicate that oxidative stress may have contributed to the development of urinary cancer induced by OTD.

## 1. Introduction

Urinary bladder cancer is the fourth most common cancer and eighth most common cause of cancer death among men in the United States [1]. The most common risk factors for bladder cancers are tobacco smoke, arsenic, radiation, and occupational and environmental carcinogens, such as aromatic amines (AAs) [2]. Four AAs, ortho-toluidine (OTD), 2-naphthylamine, 4-aminobiphenyl, and benzidine, are classified by the International Agency for Research on Cancer (IARC) as Group 1 occupational carcinogens [3].

Recently, ten workers at a Japanese chemical plant handling AAs were diagnosed with bladder cancer [4]. All patients were exposed primarily to OTD and co-exposed to *p*-toluidine, *o*-anisidine, aniline, 2,4-xylidine, acetoacet-*o*-toluidine (AAOT), or *o*-chloroaniline. Thus, OTD was considered as the causative agent of occupational bladder cancer in this plant. In our previous study, we reported that AAOT, produced using OTD as a raw material, can promote urinary bladder carcinogenesis in rats [5]. In another study, we demonstrated high concentrations of OTD in the urine of AAOT-treated rats. These data suggest that OTD, a human bladder carcinogen [6] metabolized from AAOT, can play a pivotal role in bladder carcinogenesis; thus, metabolism is an important factor for analyzing carcinogenesis. The formation of DNA adducts might be crucial in the development of OTD-related bladder carcinogenesis. However, the formation of OTD-DNA adducts remains controversial due to the absence of adducts in previous in vivo studies [7,8,9].

Comprehensive analyses of DNA adducts (called DNA adductome) using High Resolution Accurate Mass (HRAM) instruments have been performed to screen for multiple DNA adducts [10,11,12]. HRAM can acquire spectral data with precise mass measurement; therefore, it is sufficient to determine the molecular formula of an ion. Additionally, MS/MS fragmentation data can be used to detect a DNA adduct and provide structural information or confirmation. Thus, the HRAM–adductome approach is suitable for targeted analyses of previously reported DNA adducts with known chemical structures and untargeted analyses investigating unknown DNA adducts [13]. Previously, we reported the mutagenicity mechanisms for nanosized magnetite in mouse lungs and the livers of rats treated with 1,4-dioxane using this approach [10,12]. Furthermore, we determined the environmental factors causing esophageal cancer in Cixian, an area well-known for the high incidence of this cancer type in China [14]. Therefore, the aim of the present study was to analyze DNA adduct formation by AAs (OTD and AAOT) via adductome analysis and screen a few characteristic DNA adducts produced after treatment with the AAs. The accurate *m*/*z* values and MS/MS fragmentation data suggested the role of oxidative stress in the genotoxicity induced by OTD and AAOT.

Nicotinamide adenine dinucleotide phosphate (NADPH) oxidase is known to produce intracellular reactive oxygen species (ROS) and is implicated in various signaling events, including cell growth, cell survival, and cell death [15]. Apocynin, a methoxy-substituted catechol, inhibits NADPH oxidase by blocking the association of p47phox and p67phox with gp91phox [16]. We previously reported that apocynin reduced oxidative stress in arsenite- and nicotine-treated rat urothelium [17,18]. Furthermore, apocynin suppressed carcinogenesis in the pancreas and liver of rats induced by various chemicals and prostate in a transgenic model [19,20,21].

In the current study, the effects of four AAs used in the chemical plant on the ten cases of bladder cancer in male Japanese workers at the chemical plant [4] were evaluated in the rat urothelium. In addition, the mechanisms of the development of pre-cancerous lesions in rat urothelium induced by OTD/AAOT were examined by HRAM–adductome. The effects of apocynin were evaluated to determine the roles of oxidative stress in OTD-induced rat bladder carcinogenesis.

## 2. Materials and Methods

### 2.1. Test Chemical

*o*-Toluidine hydrochloride (T536215; purity, 97%) was obtained from Toronto Research Chemicals, Toronto, ON, Canada, *p*-toluidine (PT) hydrochloride (T0303; purity, >98%) was obtained from Tokyo Chemical Industry, Tokyo, Japan, and aniline (ANL) hydrochloride (017-04092; purity, 97%) was obtained from Wako Pure Chemical Industries, Osaka, Japan. AAOT (A1020; purity, >98%) and apocynin (H0261; purity, >98%) were obtained from the Tokyo Chemical Industry. Nuclease P1 and HPLC-grade acetonitrile were purchased from Wako Pure Chemical Industries, Ltd. Phosphodiesterase I was purchased from Worthington Biochem (Lakewood, NJ, USA). Bovine spleen phosphodiesterase II, DNase I, and bacterial alkaline phosphatase Type III (*Escherichia coli*) were purchased from SigmaAldrich (St. Louis, MO, USA). All other chemicals used were of analytical grade and purchased from Wako Pure Chemical Industries, Ltd.

### 2.2. Experimental Animals

The Laboratory Animal Center of the Osaka Metropolitan University Graduate School of Medicine is accredited by the Center for the Accreditation of Laboratory Animal Care and Use (CALAC), Japan Health Sciences Foundation. All animal studies were approved by the Institutional Animal Care and Use Committee of the Osaka Metropolitan University Graduate School of Medicine and conducted in accordance with the Guidelines for Proper Conduct of Animal Experiments (Science Council of Japan, 2006), the National Research Council’s Guide for the Care and Use of Laboratory Animals and ARRIVE guidelines [22]. Five-week-old male F344 rats were obtained from Charles River Japan (Atsugi, Japan) and housed in plastic cages with hardwood chip bedding in an air-conditioned room maintained at 23 °C ± 2 °C and 55% ± 5% humidity with a 12 h light/dark cycle, basal diet (Oriental MF, Oriental Yeast Co., Tokyo, Japan), and tap water *ad libitum*.

### 2.3. Experimental Design

We have conducted two types of animal experiments, one to elucidate the mechanism of urothelial toxicity induced by exposure to AAs and another to confirm the effect of antioxidants on urothelial hyperplasia induced by OTD.

At the beginning of Experiment 1, the animals were randomly allocated to 5 groups consisting of 18 rats each (Control, untreated control; ANL, aniline-treated; PT, *p*-toluidine-treated; AAOT, acetoaceto-*o*-toluidine-treated; OTD, *o*-toluidine-treated); likewise, for Experiment 2, the animals were allocated to 5 groups comprising 6 or 12 rats each (OTD, *o*-toluidine-treated; OTD + APOL, OTD plus low dose apocynin; OTD + APOH, OTD plus high dose apocynin; Control, untreated control; APOH, high dose apocynin) based on their body weights (measured just before starting the chemical treatment).

The animals in Experiment 1 were fed diets supplemented with 0.6% ANL, 0.6% PT, 1.5% AAOT, or 0.6% OTD for 4 weeks. The dose of AAOT (1.5%) is the same as that used in the previous study [5], and the dose of OTD (0.6%) is the same as that used in other carcinogenicity studies [23]. The doses of ANL and PT were adjusted to be the same as that of OTD. However, the PT-administered rats experienced reductions in body weight after one week; hence, the dose was reduced from 0.6% to 0.3% from week 2 to the end of the experiment.

In Experiment 2, the animals were fed basal diets supplemented with 0% or 0.6% OTD and administered 0, 250, or 500 mg/L of apocynin in their drinking water in light-shielded bottles for four weeks. The dose of apocynin was set based on its daily intake of drinking water, as reported previously [11,12,13,15].

The body weights of the rats and the consumption of food and drinking water were measured weekly. At the end of week 4, the rats were weighed, sacrificed by exsanguination under deep isoflurane anesthesia, and subjected to laparotomy with excision of the bladder. The bladder mucosae were collected from 12 rats in each group in Experiment 1. Briefly, the urinary bladders were quickly excised and inverted on wooden applicator sticks. After rinsing with cold-free phosphate-buffered saline, the bladder epithelial cells were removed by swirling the inverted bladders vigorously in microcentrifuge tubes. The samples were immediately frozen in liquid nitrogen and stored at −80 °C until they were processed. Bladder tissues from the remaining six rats in Experiment 1 and all rats in Experiment 2 were infused with 10% buffered formalin and fixed in 10% formalin. The fixed organs were routinely processed for sectioning and histopathological and immunohistochemical examinations. The diagnosis of the urothelium we defined in this study is based on INHAND (International Harmonization of Nomenclature and Diagnostic Criteria for Lesions in Rats and Mice) [24]. Simple hyperplasia is defined as “linear uniform thickening of the lining lacking prominent outward or inward focal growth”.

### 2.4. Immunohistochemistry and TUNEL Assays

Deparaffinized bladder sections were heated in an antigen retrieval buffer (sodium citrate, pH 6.0), treated with 3% H_2_O_2_, and incubated with rabbit monoclonal Ki67 antibody (SP6, Abcam plc, Cambridge, UK) or rabbit polyclonal phospho-H2AX (Ser139; γ-H2AX) antibody (#2577; Cell Signaling Technology, Danvers, MA, USA) overnight at 4 °C. Reactivity with the primary antibody was detected by incubating the sections with biotin-labeled goat anti-rabbit immunoglobulin (Ig)G, followed by treatment with the avidin-biotin-peroxidase complex (ABC kit; Vector, Burlingame, CA, USA) and 3,3′-diaminobenzidine tetrahydrochloride (DAB) solution (Agilent Technologies, Santa Clara, CA, USA). Deparaffinized bladder sections were heated in antigen retrieval buffer (sodium citrate pH 6.0) and incubated with Protein Block (ab64226; Abcam plc) for 30 min and anti-8-hydroxy-2′-deoxyguanosine (8-OHdG) antibody (N45.1; Japan Institute for the Control of Aging, Fukuroi, Japan) overnight at 4 °C to measure the oxidative damage. Reactivity with the primary antibody was detected by incubating the sections with N-Histofine Simple Stain Rat MAX PO (M) (Nichirei Biosciences Inc., Tokyo, Japan) and DAB solution (Agilent Technologies). Apoptotic cells in the urothelium were detected using the ApopTag^®^ Peroxidase In Situ Apoptosis Detection Kit (S7100) according to the manufacturer’s instructions (Merck KGaA, Darmstadt, Germany). The number of Ki67-, γ-H2AX-, 8-OHdG-, or TUNEL-labeled cells in at least 1000 urothelial cells in each bladder and kidney were counted to determine the corresponding labeling indices.

### 2.5. DNA Adductome Analysis

Urinary bladder mucosal epithelium was peeled into a 1.5 mL tube with a grated surface (Biomasher II) containing Tissue Lysis Buffer (QIAGEN, Hilden, Germany) with Proteinase K and SDS; DNA was extracted using the ordinary phenol/chloroform method as reported previously [25]. Desferroxamine (final concentration, 0.1 mM) was added to all DNA solutions to avoid the formation of oxidative adducts during the purification step. The extracted DNA was stored at −80 °C until DNA adductome analysis. Subsequently, the DNA samples (~30 µg) were enzymatically digested, as reported previously [10]. In most cases, DNA extracted from one animal was insufficient for adductome analysis; therefore, pooled samples were formed by combining 2–3 animals to ensure that N = 5 remains in each group. The enzymatic digestion conditions are as follows: DNA in 5 mM Tris-HCl buffer (pH 7.4) containing 5 mM CaCl_2_ employing DNase I (Type IV from bovine pancreas; 458 U) for 3 h at 37 °C. Next, nuclease P1 (from *Penicillium citrinum*; 4.8 mU), sodium acetate (pH 5.3, final 12 mM), ZnCl_2_ (final 40 mM), and HCl (final 3 mM) were added, and incubated further for 3 h at 37 °C. Alkaline phosphatase (from *E. coli*; 0.3 U), phosphodiesterase I (from *Crotalus adamanteus* venom; 0.04 U) and Tris base (final 15.4 mM) were added in the last, for an additional 16 h at 37 °C. The sample was purified using Vivacon500^®^ (10 kDa molecular weight cut-off filters, Sartorius AG, Goettingen, Germany); subsequently, the reaction mixture was centrifuged (4 °C, 10,000× *g*, 15 min) using Ultrafree^®^ (0.2 μm pore; Millipore Co., Ltd., Danvers, MA, USA) and the filtrate was used for DNA adductome analysis. Liquid chromatography (LC)–HRAM analyses were performed using the Shimadzu Prominence LC system (Kyoto, Japan) interfaced with a Triple TOF6600 mass spectrometer (SCIEX, Framingham, MA, USA) utilizing the Information Dependent Acquisition Scanning mode. The HPLC conditions were as follows: column = Synergi^TM^ Fusion-RP (2.5 μm particle size, 2.0 mm × 100 mm; Phenomenex, Torrance, CA, USA); flow rate = 0.4 mL/min; and solvent system = a linear gradient from 2.5% to 85% acetonitrile in 10 mM ammonium acetate (pH 5.3) over 30 min, controlled using the Analyst TF 1.7.1 software [10]. The sample injection volumes were 25 μL each. The MS parameters were as follows: mass range scanned from 50 to 1000 with a scan duration of 0.5 s (1.0 s total duty cycle); capillary, 3.7 kV; sampling cone, 40 V; extraction cone, 4 V; source temperature, 125 °C; and desolvation temperature, 250 °C. Nitrogen gas was used as the desolvation (flow, 800 L/h) and cone (flow, 30 L/h) gas. All data were collected using the positive ion mode, and a cone voltage of 20 V was used.

The raw data files obtained from the LC–HRAM runs were analyzed using the PeakView^®^ 2.1 and MarkerView 1.3 software (SCIEX). These applications detect, integrate, and normalize the intensities of the peaks to the sum of peaks within a specific sample. The resulting multivariate dataset, which consisted of the peak number (based on the retention time and *m*/*z*), sample name, and normalized peak intensity, was analyzed using the principal component analysis–discriminant analysis (PCA–DA).

To determine the chemical structures of the DNA adducts that were specific to OTD and/or AAOT treatment, we compared the *m*/*z* [M + H]^+^ values of the detected DNA adducts with those of known DNA adducts listed in the in-house database, as described previously [14].

### 2.6. Statistics

Statistical analyses were performed using Prism 10 (GraphPad Software, Inc., San Diego, CA, USA) and described as the mean ± standard deviation (S.D.). In Experiment 1, the homogeneity of variance was tested using the F-test. Differences in mean values between the control and aromatic amine-treatment groups were evaluated by Student’s *t*-test when the variance was homogeneous and Welch’s *t*-test when it was heterogeneous (two-group comparisons). Differences in the incidences of histopathological lesions between the control and aromatic amine-treated groups were evaluated using the two-tailed Fisher’s exact test. In Experiment 2, one-way analysis of variance and Dunnett’s or Tukey’s multiple comparisons tests were utilized. Differences in the incidences of histopathological lesions between OTD and OTD plus apocynin-treated groups were evaluated using the two-tailed Fisher’s exact test. *p* values less than 0.05 were considered significant. Intensity data obtained from the adductome analysis were expressed as the mean ± S.D. and compared with those of the corresponding solvent control/ANL/PTD using the F-test followed by Student’s *t*-test.

## 3. Results

### 3.1. Experiment 1

#### 3.1.1. Body and Organ Weights and Consumption of Food and Water

In Experiment 1, the average body weight of the PT-treated rats decreased from 120.9 g to 114.4 g after one week of treatment; therefore, the dose was reduced from 0.6% to 0.3% from week 2 through week 4. The final body weights of the rats that received all the AAs were significantly lower than those of the controls (Table 1). The absolute liver weights in the PT- and AAOT-treated rats and relative liver weights in all the AA-treated rats were significantly increased compared to those of the controls (Table 1). The food consumption in the PT- and OTD-treated and water consumption in the PT-treated were significantly lower than those in the controls (Table 1).

#### 3.1.2. Histological and Immunohistological Analyses of the Urothelium

In Experiment 1, simple hyperplasia was noted in one out of six PT-treated rats, five out of six AAOT-treated rats, and all the OTD-treated rats (Figure 1; Table 2).

No histopathological changes were observed in the ANL group (Table 2). The labeling indices of Ki67 and γ-H2AX in the urinary bladders of rats treated with AAOT and OTD were significantly higher than those of the controls (Table 2). No significant differences in TUNEL positivity were observed in the urothelium among the various groups of rats (Table 2).

#### 3.1.3. Comprehensive Analysis of DNA Adducts Induced by OTD and AAOT Treatment

The mechanisms involved in the development of pre-cancerous lesions in the rat urinary epithelia after OTD and AAOT administration were determined via adductome analysis, as described previously [12,14]. Approximately 140 types of DNA adducts were detected in the control and AA derivative-treated groups. Results of the PCA–DA against a subset of DNA adducts observed in these data sets are shown in the 2D PCA–DA score plot (Figure 2A) and associated loading plot (Figure 2B).

Clear clustering of OTD, AAOT, and others, including ANL, PT, and control, were observed (Figure 2A). This finding corresponded with the morphological changes in the rat urinary epithelium (Figure 1), suggesting that DNA adduct formation might contribute to the development of pre-cancerous lesions in the rat urinary epithelium. The associated loading plots demonstrated that the number of DNA adducts made a greater contribution to the OTD and AAOT treatments based on their PCA significance (Figure 2B). The intensity of each DNA adduct plotted in the square region was confirmed to identify the characteristic DNA adducts related to OTD and AAOT treatment (Figure 2B). Four DNA adducts were screened as candidates for OTD and AAOT exposure because their intensities were significantly higher in the OTD- and AAOT-treated groups than in the other groups.

Figure 3 shows the intensities of the four candidate DNA adducts in the control and treated groups. The DNA adduct, named adduct_557, showed significantly higher contributions following OTD and AAOT exposure. In addition, the adducts named adduct_32 and adduct_489 showed significantly higher contributions following OTD exposure. Similarly, adduct_242 showed an increasing tendency in the OTD and AAOT groups, but the results were not statistically significant.

#### 3.1.4. Identification of DNA Adducts Correlated with OTD and AAOT Treatment Using an In-House DNA Adduct Database

Our in-house DNA adduct database was used to identify the chemical structures of DNA adducts characteristic of OTD and AAOT exposure [14]. First, the *m*/*z* [M + H]^+^ values were compared with those of known DNA adducts listed in our in-house database. The *m*/*z* values of adduct_32 (*m*/*z* [M + H]^+^ 268.1051), adduct_242 (*m*/*z* [M + H]^+^ 284.1001), and adduct_489 (*m*/*z* [M + H]^+^ 244.0933) were nearly identical to those of 8-OH-dA (*m*/*z* [M + H]^+^ 268.1046), 8-OH-dG (*m*/*z* [M + H]^+^ 284.0994), and 5-OH-dC (*m*/*z* [M + H]^+^ 244.0933), respectively. Fragment ion peaks corresponding to adenine (*m*/*z* [A + H]^+^ 136.0606), guanine (*m*/*z* [G + H]^+^ 152.0566), and cytosine (*m*/*z* [C + H]^+^ 112.0504) were also observed in the MS/MS fragmentation data (Figure 4); therefore, it was concluded that these adducts were related to oxidative stress.

On the other hand, adduct_557 (*m*/*z* [M + H]^+^ 507.1582) was not found in the database and was presumed to be a novel adduct. Two fragment ion peaks corresponding to the loss of deoxyribose moiety (−116.0467) from the precursor ion peak (*m*/*z* 507.1582) and the daughter ion peak (*m*/*z* 391.1090) were observed (Figure 4). In addition, an ion peak with *m*/*z* [M + H]^+^ 127.0493 was found to be nearly identical to that of thymine moiety (*m*/*z* [M + H]^+^ 127.0515). Based on these fragment data, it was suggested that the chemical structure of adduct_557 might be inter-strand cross-linked with the thymidine base.

### 3.2. Experiment 2

Based on the adductome analysis, the levels of oxidative stress-related DNA adducts were increased in the urinary bladder of rats treated with OTD and AAOT, indicating that oxidative stress might have contributed to the formation of the pre-cancerous lesions in the rat urinary epithelia. Consequently, another animal experiment (Experiment 2) determining the effect of antioxidants on urinary epithelial hyperplasia induced by OTD was conducted to confirm this hypothesis.

#### 3.2.1. Body and Organ Weights and Consumption of Food and Water

In Experiment 2, the final body weights of rats receiving OTD, with or without apocynin treatment, were significantly lower than those of the controls (Table 3). The relative liver weights in the OTD-treated rats, with or without apocynin treatment, were significantly increased compared to those of the controls (Table 3). There were no apparent differences in food and water consumption among the groups (Table 3). Apocynin had no significant effects on the body and liver weights and the food and water consumption in the rats (Table 3).

#### 3.2.2. Histological and Immunohistological Analyses of the Urothelium

Simple hyperplasia was observed in all rats treated with OTD, all rats treated with OTD plus low dose apocynin (OTD + APOL), and in 11 of the 12 rats treated with OTD plus high dose apocynin (OTD + APOH; Figure 5, Table 4).

Apocynin treatment reduced moderate simple hyperplasia, but not significantly (*p* = 0.089). No histopathological changes were observed in the high dose apocynin (APOH) group (Table 4). The labeling indices of Ki67, γ-H2AX, and 8-OHdG in the urinary bladders of rats treated with OTD were significantly higher than those in the controls (Table 4). A significant dose-dependent decrease in the labeling indices of Ki67, γ-H2AX, and 8-OH-dG was observed in the bladder urothelium of the OTD/apocynin-treated rats (Table 4). However, no significant differences in TUNEL positivity were seen in the urothelium of rats among the various groups (Table 4).

## 4. Discussion

AAs, such as OTD, are known to cause occupational bladder cancer [3]. Recently, occupational bladder cancers were reported in male workers in a Japanese chemical plant producing organic dye and pigment intermediates. All patients from this chemical plant were primarily exposed to OTD with co-exposure to aromatic amine derivatives such as PT, *o*-anisidine, ANL, 2,4-xylidine, AAOT, or *o*-chloroaniline [4]. In the present study, short-term urinary bladder toxicity was induced in rats fed a diet containing various concentrations of AAs. Both OTD and AAOT induced simple hyperplasia and increased the number of Ki67- and γ-H2AX-positive cells in the urinary bladders; alternatively, treatment with ANL and PT did not result in any pre-cancerous lesions. The effects of AAOT and OTD in the urothelium of rats were reported in our previous studies [26,27]. Although OTD is considered a bladder carcinogen, the carcinogenic mechanisms involved remain poorly understood. Furthermore, the mutagenic activity of OTD is controversial. In several studies, including ours, OTD did not exhibit mutagenicity in the Ames salmonella assay with or without metabolic activation systems [3,28,29]. However, Beyerbach et al. reported that the 2′-deoxyguanosine-8-yl-OTD (2-methylaniline) adduct could be chemically synthesized [7]. In a recent report on the in vivo formation of the OTD-DNA adduct, released OTD was detected from DNA samples extracted from human bladder tissue after acidic hydrolysis [30]. Thus, the OTD-DNA adduct, such as dG-C8-OTD, has not been detected in biological samples so far. The dG- and dA-OTD adducts were not detected in both rat liver and human bladder samples using the LC–MS/MS instrument [8,9]. Based on this background, we conducted a comprehensive analysis of the DNA adductome to assess the carcinogenic mechanisms of OTD using animal tissue samples after exposure to AAs. Searching for the *m*/*z* [M + H]^+^ 373.1624, which corresponded to the *m*/*z* value of dG-C8-OTD, yielded no findings. This result was consistent with previous reports and suggested that dG-C8-OTD was not formed in rat bladder tissues under these conditions. In contrast, DNA adducts related to oxidative stress were identified as the characteristic DNA adducts for OTD/AAOT treatment. According to previous reports, OTD may not occur through direct DNA adduct formation after metabolic activation, but rather through their *p*-aminophenol metabolites, which undergo redox cycling via their quinone imine intermediates, producing ROS as byproducts that can damage DNA and induce mutations [31,32]. Ascorbic acid, which scavenges ROS, was reported to protect the genetic and epigenetic alterations induced by 3,5-dimethylaminophenol in vitro [33]. Thus, the possible mechanisms of OTD genotoxicity are shown in Figure 6. The *p*-aminophenol metabolites of OTD, *N*-acethyl-4-amino-*m*-cresol, are thought to form protein adducts in urinary epithelial cells [29], which might stimulate immune cells and possibly induce inflammation along with ROS production via NADPH oxidase. This hypothesis was tested by conducting an animal experiment, which confirmed the effect of the NADPH oxidase inhibitor, apocynin, on urinary epithelial hyperplasia in rats induced by OTD (Figure 5). Moreover, the numbers of Ki67-, γ-H2AX-, and 8-OH-dG-positive cells in the urinary bladder of rats administered apocynin were significantly decreased compared to those in the controls (Table 4). Thus, it is likely that oxidative stress contributed to the development of urinary bladder cancer after exposure to OTD.

Previously, we demonstrated the positive effect of AAOT on BBN-induced urinary bladder carcinogenesis in rats due to the overexpression of JUN, a transcription factor in the activator protein-1 (AP-1) complex, and its downstream target genes [5]. In addition to the promotion effects, AAOT induced the development of pre-cancerous lesions, such as hyperplasia and γ-H2AX positivity in the urinary bladder epithelium. γ-H2AX is a biomarker of DNA damage and is useful for predicting urinary bladder cancer [34]. Therefore, AAOT is speculated as a causative agent of urinary bladder cancer, both as a promoter and initiator. We have previously reported high concentrations of OTD in the urine of AAOT-treated rats [5]; hence, we hypothesized that the carcinogenic mechanism of AAOT is the OTD metabolically produced from AAOT. The extent of hyperplasia and the number of Ki67- and γ-H2AX-positive cells in urinary bladders induced by AAOT seemed slightly weaker than those seen in bladders treated by OTD (Figure 1; Table 2). Likewise, a clear clustering of the adductome data was not observed in the AAOT-treated group compared to that in the OTD-treated group (Figure 2A). Mechanisms other than oxidative stress could be involved in the urinary bladder carcinogenesis induced by AAOT/OTD. In this study, presumed cross-linked adducts (adduct_557) have also been detected. It is possible that these bulky adducts are responsible for the inability of antioxidants to decrease DNA damage as measured by γ-H2AX. In our previous study, OTD exhibited co-mutagenesis activity with non-mutagenic β-carbolines, such as norharman, and produced the potent mutagenic compound 9-(4′-amino-3′-methylphenyl)-9*H*-pyrido [3,4-*b*]indole (aminomethylphenylnorharman; AMPNH) [35]. AMPNH can be produced in the body because norharman is an endogenous compound. In another study, γ-H2AX formation was significantly increased in the bladder epithelial cells of rats following AMPNH treatment compared to that in controls, thus indicating that AMPNH may be a potent causative agent of bladder carcinogenesis [36].

## 5. Conclusions

In conclusion, OTD-related AAs (AAOT and OTD) have the potential to significantly induce simple hyperplasia, cell proliferation, and DNA damage in the rat urothelium. DNA adductome analysis showed that oxidative stress, probably produced via the metabolic activation of OTD, may have contributed to the development of urinary cancer in this study. However, it is not known whether the mechanisms involved in occupational bladder cancer are similar to those in rats. An integrated approach that combines data from adductome analysis, next-generation sequencing, and animal experiments would be crucial to identify the causative agents and mechanisms involved in occupational bladder cancer. Currently, we are conducting the whole genome sequencing of urinary epithelia from rats fed with OTD using error-corrected sequencing technology to obtain the OTD-related mutational signature. Simultaneously, whole genome sequencing of occupational bladder cancers is also underway. The information collected by these experiments will help elucidate the relationship between oxidative stress and occupational bladder cancer.

## Figures and Tables

**Figure 1 biomolecules-14-00036-f001:**
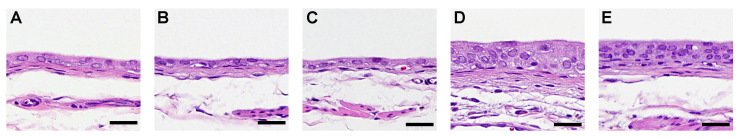
Histology of the rat urinary bladder epithelium treated with aromatic amines (AAs). Typical histopathological changes by AAs in the urinary bladder epithelium of the (**A**) untreated control, (**B**) aniline-treated rats, (**C**) *p*-toluidine-treated rats, (**D**) acetoaceto-*o*-toluidine-treated rats, and (**E**) *o*-toluidine-treated rats. Scale bar = 25 µm.

**Figure 2 biomolecules-14-00036-f002:**
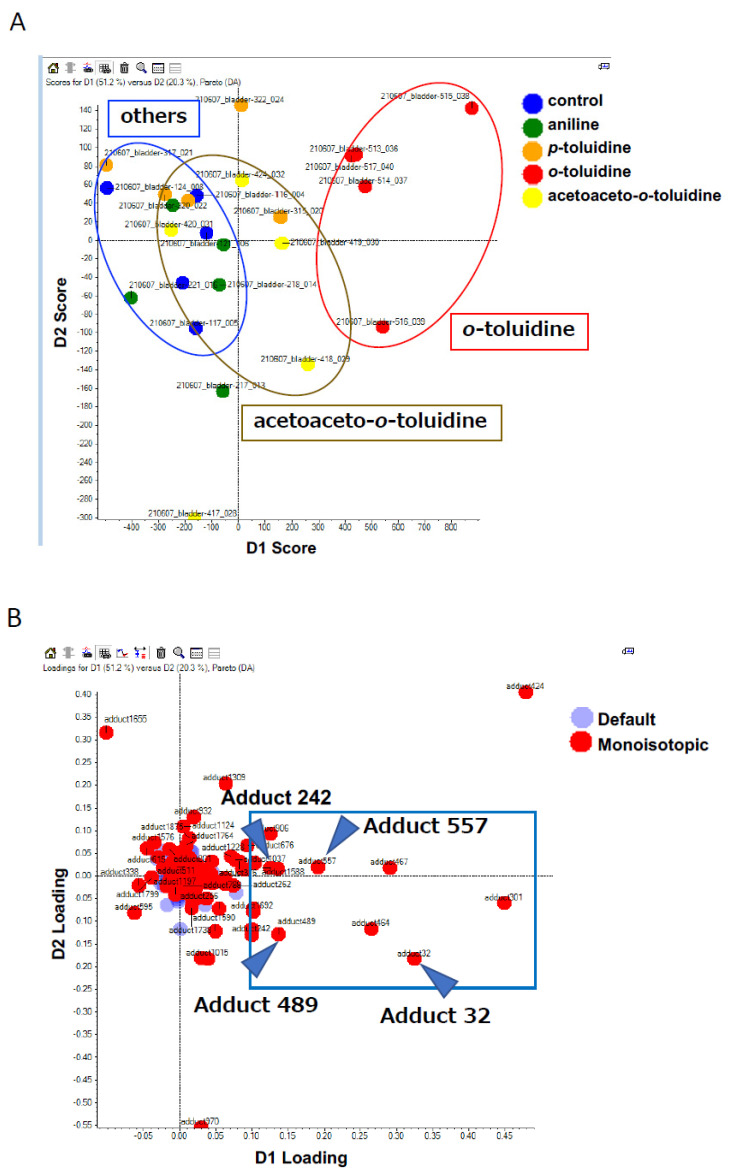
PCA−DA scores and loading plots. (**A**) 2D PCA–DA scores of DNA adducts obtained from the adductome analysis. Blue, control; red, OTD; yellow, AAOT; orange, PT; and green, ANL. The clusters are indicated by ellipses: red for OTD, yellow for AAOT, and blue for others. (**B**) Variable loading plots. Each red spot represents DNA adducts observed in the DNA adductome analysis. The intensity of each DNA adduct enclosed in a square region was confirmed. Arrowheads indicate the candidate characteristic DNA adducts for OTD/AAOT exposure.

**Figure 3 biomolecules-14-00036-f003:**
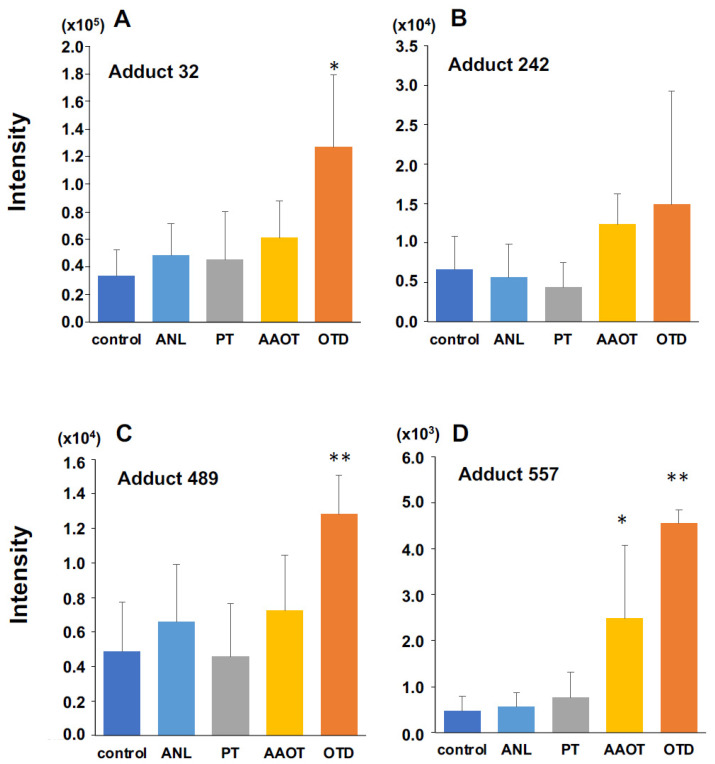
Quantitative analysis of three DNA adducts screened as characteristic DNA adducts for OTD/AAOT treatment. The intensity of DNA adducts (**A**) adduct 32, (**B**) adduct 242, (**C**) adduct 489, and (**D**) adduct 557 screened as characteristic DNA adducts for OTD/AAOT treatment. * *p* < 0.05, ** *p* < 0.01, Student’s *t*-test.

**Figure 4 biomolecules-14-00036-f004:**
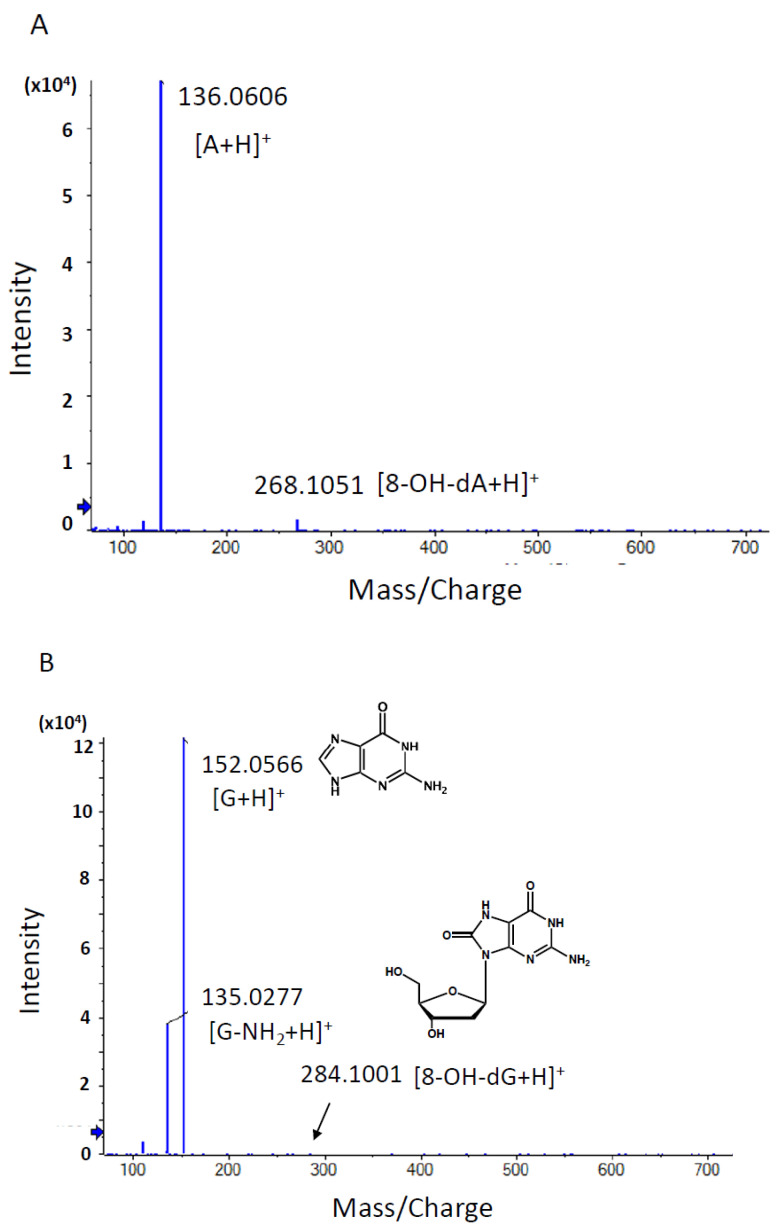
Product ion scan and MS/MS fragmentation data of characteristic DNA adducts. Using HRAM data, a candidate formula of the adduct was automatically calculated using the PeakView^®^ software (SCIEX), as described above. (**A**) MS/MS fragmentation data of adduct 32 with a retention time of 2.04 min. The *m*/*z* [M + H]^+^ value of the precursor ion was nearly identical to that of 8-OH-dA (*m*/*z* [M + H]^+^ 268.1046), and a fragment ion with *m*/*z* 136.0606 [M + H]^+^ showing a similar *m*/*z* value to adenine with a value of *m*/*z* 136.0630 [A + H]^+^ was observed. (**B**) MS/MS fragmentation data of adduct 242 with a retention time of 2.19 min. The *m*/*z* [M + H]^+^ value of the precursor ion was nearly identical to that of 8-OH-dG (*m*/*z* [M + H]^+^ 284.0994), and a fragment ion with *m*/*z* 152.0566 [M + H]^+^ showing a similar *m*/*z* value to guanine with a value of *m*/*z* 152.0580 [G + H]^+^ was observed. (**C**) MS/MS fragmentation data of adduct 489 with a retention time of 1.76 min. The *m*/*z* [M + H]^+^ value of the precursor ion was nearly identical to that of 5-OH-dC (*m*/*z* [M + H]^+^ 244.0933), and a fragment ion with *m*/*z* 112.0504 [M + H]^+^ showing a similar *m*/*z* value to cytosine with a value of *m*/*z* 112.0518 [C + H]^+^ was observed. (**D**) MS/MS fragmentation data of adduct 557 with a retention time of 2.67 min. A peak corresponding to the loss of dR moiety (−116.0467) from the precursor ion peak *(m*/*z* 290.0860) and daughter ion peak (*m*/*z* 391.1090) was observed. An ion peak with *m*/*z* [M + H]^+^ 127.0493 was nearly identical to that of thymine moiety (*m*/*z* [M + H]^+^ 127.0515).

**Figure 5 biomolecules-14-00036-f005:**
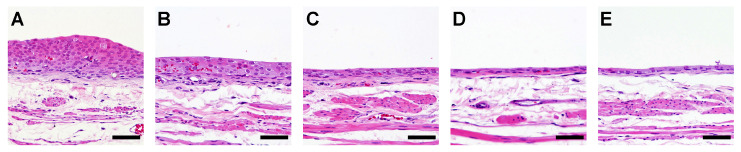
Histology of the rat urinary bladder epithelium treated with OTD and/or apocynin. Typical and specific histopathological changes by OTD and/or apocynin in the urinary bladder epithelia of rats from groups (**A**) treated with OTD, (**B**) treated with OTD + low dose apocynin, (**C**) treated with OTD + high dose apocynin, (**D**) Untreated control, and (**E**) treated with high dose apocynin. Moderate and mild simple hyperplasia were detected in groups (**A**) and (**B**), respectively. One rat in group (**C**) showed no hyperplasia. Scale bar = 50 µm.

**Figure 6 biomolecules-14-00036-f006:**
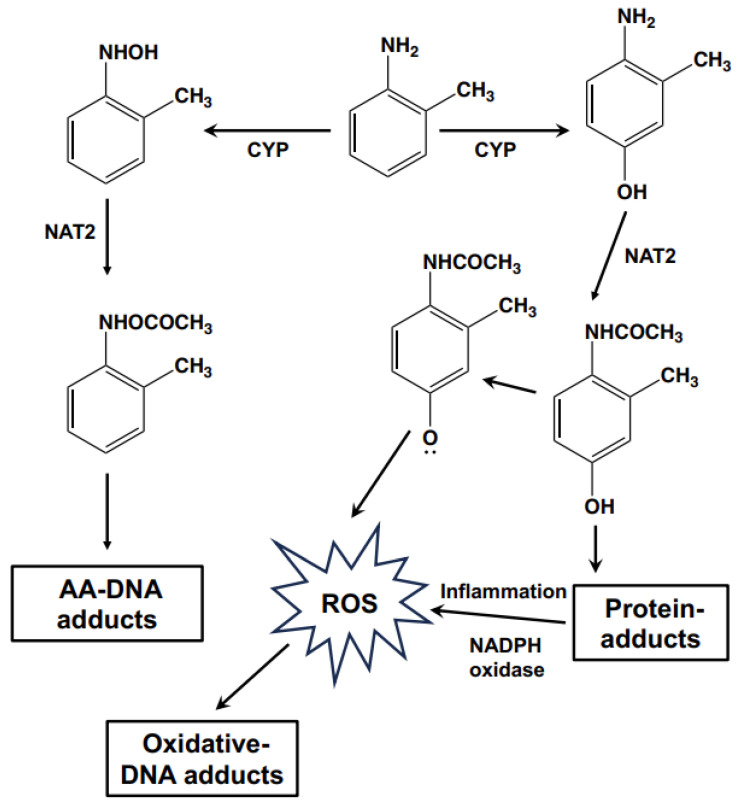
Schematic illustration of the possible genotoxic mechanisms of OTD.

**Table 1 biomolecules-14-00036-t001:** Body weight, liver weight, and food and water consumption in animal experiment 1.

Treatment	No. of Rats	Body Weight (g)			Liver (No. = 6)						Consumption					
					Absolute (g)			Relative (%)			Food (g/day)			Water (g/day)		
Control	18	228.2	±	8.3	8.1	±	0.3	3.5	±	0.1	13.2	±	0.8	19.7	±	1.1
ANL	18	214.7	±	6.5 ***	8.5	±	0.2	3.9	±	0.0 ***	12.3	±	1.5	20.2	±	1.6
PT	18	200.2	±	8.0 ***	11.2	±	0.7 ***	5.5	±	0.2 ***	10.7	±	3.4 ***	18.0	±	3.4 *
AAOT	18	207.5	±	10.1 ***	9.3	±	0.7 **	4.4	±	0.2 ***	12.8	±	1.1	19.1	±	1.3
OTD	18	210.2	±	8.2 ***	8.6	±	0.6	4.1	±	0.2 ***	11.8	±	1.9 *	19.3	±	2.0

*, **, *** represent *p* < 0.05, 0.01, and 0.001 vs. control, respectively. ANL: aniline; PT: *p*-toluidine; AAOT: acetoaceto-*o*-toluidide; OTD: *o*-toluidide.

**Table 2 biomolecules-14-00036-t002:** The effects of aromatic amines on urinary bladder in animal experiment 1.

Treatment	No. of Rats	Simple Hyperplasia	Ki67 (%)			γH2AX (%)			TUNEL (%)		
Control	6	0	1.7	±	0.4	0.6	±	0.2	0.6	±	0.4
ANL	6	0	2.0	±	0.5	0.9	±	0.4	0.4	±	0.3
PT	6	1	1.7	±	0.4	0.7	±	0.3	0.4	±	0.2
AAOT	6	5 *	3.6	±	0.7 ***	1.9	±	0.7 ***	0.5	±	0.3
OTD	6	6 **	4.9	±	1.3 ***	2.6	±	0.8 ***	0.6	±	0.2

*, **, *** represent *p* < 0.05, 0.01, and 0.001 vs. control, respectively. ANL: aniline; PT: *p*-toluidine; AAOT: acetoaceto-*o*-toluidide; OTD: *o*-toluidide.

**Table 3 biomolecules-14-00036-t003:** Body, liver weight, and food and water consumption in animal experiment 2.

Treatment	No. of Rats	Body Weight (g)			Liver (No. = 6)						Consumption					
					Absolute (g)			Relative (%)			Food (g/day)			Water (g/day)		
OTD	12	215.1	±	5.9 **	8.9	±	0.5	4.1	±	0.2 ***	11.5	±	2.0	20.1	±	2.0
OTD + APOL	12	214.1	±	8.9 ***	9.0	±	0.7	4.1	±	0.1 ***	11.8	±	1.7	20.5	±	1.8
OTD + APOH	12	213.1	±	6.7 ***	9.0	±	0.2	4.2	±	0.1 ***	11.7	±	1.9	19.1	±	2.0
Control	6	232.0	±	11.3	8.5	±	0.5	3.7	±	0.1	12.9	±	1.4	20.1	±	1.0
APOH	6	237.2	±	8.5	8.8	±	0.3	3.7	±	0.0	13.3	±	1.3	20.1	±	0.6

**, *** represent *p* < 0.01 and 0.001 vs. control, respectively. OTD: *o*-toluidide; APO: apocynin; L: low dose (250 mg/L); H: high dose (500 mg/L).

**Table 4 biomolecules-14-00036-t004:** The effects of OTD and apocynin on urinary bladder urothelium in animal experiment 2.

Treatment	No. of Rats	Normal	Simple Hyperplasia		Ki67 (%)			γ-H2AX (%)			TUNEL (%)			8-OHdG (%)		
			Mild	Moderate												
OTD	12	0	2	10	2.1	±	0.7 ^a^	1.7	±	0.7 ^a^	0.7	±	0.3	1.7	±	0.3 ^a^
OTD + APOL	12	0	3	9	1.5	±	0.9	1.0	±	0.4 **	0.6	±	0.3	1.2	±	0.2 ***
OTD + APOH	12	1	6	5	0.9	±	0.4 ***	0.7	±	0.3 ***	0.5	±	0.2	0.9	±	0.2 ***
Control	6	6	0	0	0.7	±	0.4	0.2	±	0.1	0.7	±	0.6	0.3	±	0.1
APOH	6	6	0	0	0.6	±	0.4	0.3	±	0.1	0.6	±	0.6	0.3	±	0.1

^a^: *p* < 0.001 vs. control; **, *** represent *p* < 0.01 and 0.001 vs. OTD, respectively. OTD: *o*-toluidide; APO: apocynin; L: low dose (250 mg/L); H: high dose (500 mg/L).

## Data Availability

The data presented in this study are available in the article.

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
