# Peer review of "Evaluation of the Mechanisms Involved in the Development of Bladder Toxicity following Exposure to Occupational Bladder Cancer Causative Chemicals Using DNA Adductome Analysis"

_biomolecules, 2023, doi:10.3390/biom14010036_

Round 1

Reviewer 1 Report

Comments and Suggestions for Authors

This is a well-controlled study employing a rat model to investigate the relationship between aromatic amine (AA) exposure and bladder-tissue neoplasia. What might otherwise have been a simple descriptive analysis is considerably augmented and strengthened through detailed analysis of the exposed bladder tissue’s DNA-adduct landscape. This “adductome” analysis is a relatively new method for characterizing and quantifying the presence of covalently modified nucleosides and DNA bases in genomic DNA. Based on these adductome data, the authors propose a pathway for AA-dependent bladder-epithelium mutagenesis promoted by reactive-oxygen species.

I found little to criticize about this manuscript except for one minor editorial suggestion. A more detailed description of the adductome analysis would be beneficial for most readers, especially regarding details such as genomic-DNA preparation. Citation of reference [10] is not optimal here as that paper does not offer significantly greater detail than the present manuscript.

Finally, please increase the font size of axis labels and annotations in Figure 2.

Reviewer 2 Report

Comments and Suggestions for Authors

This study provided valuable data for better understanding the genotoxicity of acetoacet-o-toluidine (AAOT) and o-toluidine (OTD), via adductome analysis and experiments for proving the hypothesis of oxidative stress-induced genotoxicity.

I have some questions and suggestion for discussion:

1. What is the dose design evidence in rat experiments? If there are reference please provide them in background, compared with LD50?

2. Have you checked the OTD tissue concentration after treatment of AAOT? since OTD might be the metabolite of AAOT, then the comparison can be done between the two groups, OTD and AAOT treated groups.

3. How about the adducts level changes in the experiment 2, after the oxidative stress suppressed by apocynin, have you conducted the detection in experiment 2? Did -OH dA/G/C decreased? And the most interested one adduct-557, will it be decreased by inhibition of ROS?

4. I wonder the contribution of oxidative stress in OTD/AAOT-induced genotoxicity, because the damage endpoints of r-H2AX and others seems not be attenuated completely. How about the determination of DNA repair pathway? Since the ROS damage should be repaired by different enzyme compared with bulky DNA adducts, as authors mentioned the main adduct-557 might be from inter-strand cross-linked with the thymidine base. 

Reviewer 3 Report

Comments and Suggestions for Authors

1. Please add descriptive diagnostic names of Figure 1A to E and Figure 5A to E. Define "simple hyperplasia" for unfamiliar readers. Are there "complex" hyperplasia or "Atypical" hyperplasia like in some of human pathology jargons? All of them seem to have some atypical (larger nuclei) cells more or less.

2. The authors show 4 adducts with m/z and retention times, but without annotation. Line 5 in the abstract should be clearly presented like "PCA analysis of numerous candidate adducts clearly categorized adducts generated from OTD and AAOT treatment from those in controls". The adductome results  are embedded while title include adductomics as a keyword. The results shown in Line 280 to 284 should be included in the abstract.

3. Investigation of pathological markers Ki67 (proliferation), γ-H2AX (DNA damage), and 8-OHdG (oxidative stress) are sound. Counting 1000 cells is painstaking. How many slides do the authors use?

4. When 45.1 was used, did the authors absorb with urine? That antibody reacts with urea (Kasai, personal information) in some settings. Does it recognize 8-OHdA ?
